# PAY ATTENTION TO CYCLE FOR SPATIO-TEMPORAL GRAPH NEURAL NETWORK

## ABSTRACT

Graph Neural Networks (GNNs) and Transformer have been increasingly adopted to learn the complex vector representations of spatio-temporal graphs, capturing intricate spatio-temporal dependencies crucial for applications such as traffic datasets. Although many existing methods utilize multi-head attention mechanisms and message-passing neural networks (MPNNs) to capture both spatial and temporal relations, these approaches encode temporal and spatial relations independently, and reflect the graph's topological characteristics in a limited manner. In this work, we introduce the Cycle to Mixer (Cy2Mixer), a novel spatio-temporal GNN based on topological non-trivial invariants of spatio-temporal graphs with gated multi-layer perceptrons (gMLP). The Cy2Mixer is composed of three blocks based on MLPs: A message-passing block for encapsulating spatial information, a cycle message-passing block for enriching topological information through cyclic subgraphs, and a temporal block for capturing temporal properties. We bolster the effectiveness of Cy2Mixer with mathematical evidence emphasizing that our cycle message-passing block is capable of offering differentiated information to the deep learning model compared to the message-passing block. Furthermore, empirical evaluations substantiate the efficacy of the Cy2Mixer, demonstrating state-of-the-art performances across various traffic benchmark datasets.

## 1 INTRODUCTION

Traffic forecasting aims to predict future traffic in road networks based on preceding traffic data (Li et al., 2017; Deng et al., 2021; Shao et al., 2022a;b; Zhu et al., 2023). In Intelligent Transportation Systems (ITS), traffic forecasting has a pivotal role in effective urban management (Vlahogianni et al., 2004; Yin et al., 2015) due to its potential to benefit a wide range of applications related to road networks, including route planning, vehicle dispatching, and congestion mitigation (Wang et al., 2021). Traffic data is frequently conceptualized as a spatio-temporal graph, where road connections and traffic flows gleaned from sensors are represented as edges and nodes (Diao et al., 2019; Wang et al., 2020b). Given this, many turn to Graph Neural Networks (GNNs), especially those using Message Passing Neural Networks (MPNNs), to study this data. GNNs based on MPNNs have been increasingly adopted to learn the complex vector representations of spatio-temporal graphs, capturing intricate dependencies in traffic datasets (Yu et al., 2017; Li & Zhu, 2021; Choi et al., 2022a). However, traditional message passing-based methodologies have exhibited limitations, including the inability to adequately account for temporal variations in traffic data and over-smoothing problems (Jiang et al., 2023). Following the success of Transformers in various domains, they have been employed either independently or synergetically with MPNNs to address these challenges in traffic forecasting (Zheng et al., 2020; Xu et al., 2020; Jiang et al., 2023). In some cutting-edge implementations, they are even coupled with preprocessing techniques like Dynamic Time Warping (DTW) (Berndt & Clifford, 1994), showcasing competitive performance outcomes (Jiang et al., 2023).

Previous research efforts have predominantly focused on employing additional algorithms or incorporating intricate structures to capture the complex patterns inherent in spatio-temporal graphs. This often results in heuristics that inflate the complexity of structure and computational costs. Additionally, many of these studies have leaned heavily on experimental results to justify their performance improvements, lacking comprehensive explanations or theoretical grounding. To address these challenges, we start with a mathematical hypothesis that leverages the topological non-trivial invariants

of a spatio-temporal graph to enhance the predictive performances of GNNs. This hypothesis aims to discern the spatio-temporal relationships among nodes within the graph. We demonstrate that topological features shed light on facets of temporal traffic data that might be overlooked when focusing solely on the pre-existing edges connecting nodes in a traffic network.

Building on this mathematical foundation, we propose a simple yet unique model, Cycle to Mixer (Cy2Mixer), that integrates gated Multi-Layer Perceptron (gMLP) Liu et al. (2021) and MPNNs, supplemented with topological non-trivial invariants in graphs, to enhance predictive accuracy. Our Cy2Mixer layer comprises three key components: (a) Message-passing block: Encompasses spatial relationships along with local neighborhoods. (b) Cycle message-passing block: Captures supplementary information among nodes in cyclic subgraphs. (c) Temporal block: Seizes the temporal characteristics of all nodes. Notably, our cycle message-passing block aims to encapsulate topological features, enabling a finer comprehension of the intricate connectivity patterns within the network inspired by Cy2C-GNNs (Choi et al., 2022b). We validate our methodology using real-world public traffic datasets, showcasing the competitive edge of our proposed models. Furthermore, a qualitative assessment elucidates the efficacy of topological invariants in enhancing the Cy2Mixer's predictive performance. Our work also presents an evaluation of the efficiency in preprocessing and training time of our model relative to recent state-of-the-art techniques.

The contributions of this work are as follows: (1) We use homotopy invariance of fundamental groups and homology groups to deduce that topological non-trivial invariants of a traffic network become a contributing factor for influencing the traffic forecast. (2) We propose a simple yet novel network, Cy2CMixer, based on the gMLP and Cy2C-GNNs inspired by the theory of universal covering spaces. (3) The proposed model not only exhibits minimal computational cost but also proves to be highly efficient, consistently delivering superior performance across a variety of datasets, including traffic forecasting.

## 2 PRELIMINARIES

In this section, we provide explanations for some notations and proceed to define the traffic prediction problem.

### 2.1 MESSAGE PASSING NEURAL NETWORKS (MPNNS)

Each $m$-th layer $H^{(m)}$ of the MPNNs and hidden node attributes $h_v^{(m)}$ with dimension $k_m$ are defined as :

$$\begin{cases} h_v^{(m)} := \text{COMBINE}^{(m)} \left( h_v^{(m-1)}, \text{AGGREGATE}_v^{(m)} \left( \left\{\!\!\left\{ h_u^{(m-1)} \mid u \in N(v) \right\}\!\!\right\} \right) \right) \\ h_v^{(0)} := X_v \end{cases}$$

where $X_v$ is the initial node attribute at $v$ and $N(v)$ is the set of neighborhood nodes. Note that $\text{AGGREGATE}_v^{(m)}$ is a function that aggregates features of nodes adjacency to $v$, and $\text{COMBINE}^{(m)}$ is a function which combines features of the node $v$ with those of nodes adjacent to $v$.

### 2.2 GATED MULTI-LAYER PERCEPTRON (GMLP)

gMLP has been shown to achieve comparable performance to Transformer models across diverse domains, including computer vision and natural language processing, with improved efficiency (Liu et al., 2021). With the given input $X \in \mathbb{R}^{n \times d}$, where $n$ denotes sequence length and $d$ denotes dimension, it can be defined as:

$$Z = \sigma(XU), \quad \tilde{Z} = s(Z), \quad Y = \tilde{Z}V,$$

where $\sigma$ represents the activation function, which is GeLU (Hendrycks & Gimpel, 2023) in this context, whereas $U$ and $V$ correspond to linear projections based on the channel (feature) dimensions, serving roles similar to the feed-forward network in Transformer. It is important to note that the layer $s(\cdot)$ corresponds to the Spatial Gating Unit, which is responsible for capturing cross-token interactions. We construct the aforementioned layer as follows:

$$s(Z) = Z_1 \odot f_{W,b}(Z_2),$$

where $f_{W,b}$ denotes a linear projection $f_{W,b}(Z) = WZ + b$. Here, $W$ and $b$ refer to the weight matrix and bias, respectively. $Z_1$ and $Z_2$ denote two independent components split from $Z$ along the channel dimension, and $\odot$ denotes element-wise multiplication.

## 2.3 TRAFFIC PREDICTION PROBLEM

**Traffic sensor**   Traffic sensors are deployed within the traffic system to record essential information, such as the flow of vehicles on roads and the speeds of these vehicles.

**Traffic network**   Traffic network can be represented as $\mathcal{G} = (\mathcal{V}, \mathcal{E}, A, A_C)$, where $\mathcal{V} = \{v_1, \cdots, v_N\}$ denotes the set of $N$ nodes representing sensors within the traffic network ($|V| = N$). Next, $\mathcal{E} \subseteq \mathcal{V} \times \mathcal{V}$ represents the set of edges, and the adjacency matrix $A$ of the network $\mathcal{G}$ can be obtained based on the distances between nodes. Additionally, $A_C$ is the clique adjacency matrix of the network, which is incorporated into the GNN architecture. Note that the clique adjacency matrix is obtained by using the identical procedure as in the prior study (Choi et al., 2022b). It is paramount to underscore that $A \in \mathbb{R}^{N \times N}$ and $A_C \in \mathbb{R}^{N \times N}$ are time-independent input variables since the structure remains unchanged over time.

**Traffic signal**   The traffic signal $X_t \in \mathbb{R}^{N \times C}$ represents the data measured at time $t$ across $N$ nodes in the network. In this context, $C$ denotes the number of features being recorded by the sensors, and in this study, it represents the flow of the road network.

**Problem formalization**   In traffic forecasting, our objective is to train a mapping function $f$ to predict future traffic signals by utilizing the data observed in the previous $T$ steps, which can be illustrated as follows:

$$\left[X_{(t-T+1)}, \cdots, X_t; \mathcal{G}\right] \xrightarrow{f} \left[X_{(t+1)}, \cdots, X_{(t+T')}\right].$$

## 3 MATHEMATICAL BACKGROUNDS

One of the prominent ways to mathematically quantify the effectiveness and discerning capabilities of GNNs is to interpret the graph dataset as a collection of 1-dimensional topological spaces $\mathcal{G} := \{G_i\}_i$, and identify the given GNNs as a function $GNN : \mathcal{G} \to \mathbb{R}^k$ which represents graphs in a given dataset as real $k$-dimensional vectors. These approaches were carefully executed in previous literature, which focused on pinpointing the discernability of various architectural designs for improving GNNs, as seen in Choi et al. (2022b); Bodnar et al. (2021); Horn et al. (2021); Park et al. (2022). Throughout these references, the characterizations of topological invariants of such 1-dimensional topological spaces provided grounds for verifying whether such GNNs can effectively capture geometric non-trivial properties of graph datasets, such as cyclic substructures or connected components of graphs. In light of these previous studies, it is hence of paramount interest to reinterpret temporal graph datasets as a collection of higher dimensional topological spaces and understand what non-trivial properties of such topological spaces the novel deep learning techniques processing temporal graph datasets should encapsulate.

**Topological Space**   In this work, we interpret the traffic dataset as a 2-dimensional topological space constructed from the traffic network $\mathcal{G}$, and traffic signals as a function defined over the 2-dimensional topological space. To elaborate, the previous section indicates that we can identify a traffic network with a graph $\mathcal{G} = (\mathcal{V}, \mathcal{E}, A, A_C)$, and a traffic signal at time $t$, denoted by $X_t \in \mathbb{R}^{N \times C}$, as a collection of functions $\{f_t : \mathcal{G} \to \mathbb{R}^{N \times C}\}_t$, each of which represents measurements taken over the graph $\mathcal{G}$ at time $t$. But notice that the continuous time variable $t$ parametrizes a closed interval $I := [t_s, t_e]$, where $t_s$ and $t_e$ denote the start and end time of the traffic dataset. This observation allows us to identify a temporal traffic network as a topological space $\mathcal{G} \times I$, and the collection of traffic signals varying with respect to a time variable $t$ as a function over $\mathcal{G} \times I$ satisfying the following condition:

$$X : \mathcal{G} \times I \to \mathbb{R}^{N \times C}$$
$$X(g, t) = X_t(g) \text{ for all } t \in I.$$

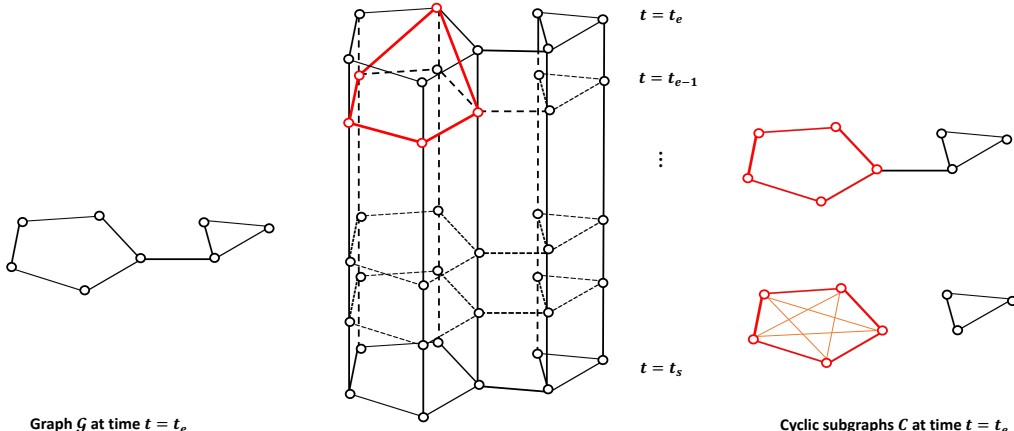

Figure 1: An illustration of lifting a cyclic subgraph of the traffic network $\mathcal{G}$ to a temporal cyclic subgraph of the topological space $\mathcal{G} \times I$ representing the traffic dataset.

**Temporal cyclic structures** One of the prominent topological properties of a graph is the configuration of its cyclic subgraphs. The existence of such cyclic subgraphs indicates that there exists a pair of nodes $v, w \in \mathcal{V}$ which are connected by at least two distinct paths, each of which is comprised of moving along the edges of the graph $\mathcal{G}$. In terms of traffic dataset, the existence of a cyclic subgraph indicates that there are at least two paths that a flow of traffic can move along from one point to another. This observation hints that it is of great importance to understand how such cyclic subgraphs of a given traffic network $\mathcal{G}$ affect predicting future traffic signals. One of the natural questions we may ask is to understand all possible cyclic substructures inherent in a traffic signal $X : \mathcal{G} \times I \to \mathbb{R}^{N \times C}$ one needs to analyze. The following result shows that cyclic substructures of a traffic signal can be fully understood by understanding the cycle bases of a traffic network $\mathcal{G}$.

**Theorem 3.1.** *Given any choice of $t_0 \in I$, let $\pi_{t_0} : \mathcal{G} \times I \to \mathcal{G} \times \{t_0\} \cong \mathcal{G}$ be the projection map which sends the interval $I$ to a singleton set $\{t_0\}$. Let $\mathcal{C}_{\mathcal{G} \times I} := \{C_1, C_2, \cdots, C_n\}$ be a cycle basis of the topological space $\mathcal{G} \times I$. Then the set*

$$\pi_{t_0} \left( \mathcal{C}_{\mathcal{G} \times I} \right) := \{\pi_{t_0}(C_1), \cdots, \pi_{t_0}(C_n)\}$$

*is a cycle basis of the traffic network $\mathcal{G}$.*

*Proof.* The theorem is a corollary of Hatcher (2002)[Proposition 1.18, Corollary 2.11]. To elaborate, the map $\pi_{t_0} : \mathcal{G} \times I \to \mathcal{G}$ is a homotopy equivalence between the space $\mathcal{G} \times I$ representing the temporal traffic data and the space $\mathcal{G}$ representing the traffic network. The map induces an isomorphism between the fundamental groups and the homology groups of two topological spaces with rational coefficients. That is, these two maps are isomorphisms of groups for all indices $i \geq 0$:

$$(\pi_{t_0})_* : \pi_1(\mathcal{G} \times I) \to \pi_1(\mathcal{G})$$

$$(\pi_{t_0})_* : H^i(\mathcal{G} \times I, \mathbb{Q}) \to H^i(\mathcal{G}, \mathbb{Q})$$

The isomorphisms between the fundamental groups $\pi_1$ and the first homology groups $H_1$ indicate that the set $\pi_{t_0}(\mathcal{C}_{\mathcal{G} \times I})$ has to be a cycle basis of $\mathcal{G}$. □

The above theorem shows in particular that any cyclic subgraph of the traffic signal $\mathcal{G} \times I$ is comprised of nodes of form $(c_1, t_1), (c_2, t_2), \cdots, (c_n, t_n) \in \mathcal{G} \times I$, where the nodes $c_1, c_2, \cdots, c_n$ are on a common cyclic subgraph of $\mathcal{G}$, and the coordinates $t_i$ correspond to different temporal instances. Theorem 3.1 mathematically demonstrates that the topological non-trivial invariants of a traffic network $\mathcal{G}$ can become a contributing factor for influencing the temporal variations measured among the nodes of traffic dataset, a potential aspect of temporal traffic data that may not be fully addressed from solely analyzing the set of pre-existing edges connecting the nodes of a traffic network. To elaborate, topological non-trivial invariants of $\mathcal{G}$ elucidate restrictions associated to constructing a global traffic signal $X : \mathcal{G} \times I \to \mathbb{R}^{N \times C}$ from coherently gluing a series of local temporal traffic signals $\{X_t\}$ measured at each time $t$. This originates from previously studied mathematical insights

that obstructions in gluing continuous functions $f_\alpha : U_\alpha \to \mathbb{R}^k$ defined over collections of open covers $U_\alpha \subset Y$ of a topological space (such that $\cup_\alpha U_\alpha = Y$) to a continuous function $f : Y \to \mathbb{R}^k$ can be detected from topological (or cohomological) invariants of the topological space $Y$ (see for example Chapter 2 or 3 of Hartshorne (1977)). Given that traffic forecasting beyond time $t = t_e$ requires a thorough understanding of traffic signals $X : \mathcal{G} \times [0, t_e] \to \mathbb{R}^{N \times C}$, we can hence conclude that a potential candidate to boost performances of traffic forecasting algorithms is to effectively incorporate topological invariants of traffic networks $\mathcal{G}$.

**Temporal cliques**     Theorem 3.1 demonstrates that every cyclic structure of temporal graph datasets $\mathcal{G} \times I$ originating from lifting the cyclic structures of the baseline traffic network $\mathcal{G}$ to any time period using the projection map $\pi_{t_0}$. In fact, the lifting procedure does not require that the cyclic structure over $\mathcal{G} \times I$ has to be defined over a fixed time $t$. Rather, the cyclic subgraph of $\mathcal{G}$ can be lifted to a cyclic subgraph of $\mathcal{G} \times I$ in a manner that the nodes on the subgraph express traffic signals measured at different moments of time.

One of the topological invariants that conventional GNNs fail to incorporate effectively is the cyclic substructures of graphs Xu et al. (2019); Bodnar et al. (2021). These unsuccessful attempts originate from the capability of GNNs to distinguish any pairs of two graphs $G$ and $H$ up to isomorphism of their universal covers or unfolding trees. One effective method to overcome these limitations is by substituting cyclic subgraphs into cliques. An approach inspired by the theory of covering spaces, the strategy alters the geometry of universal covers to allow GNNs to encapsulate cyclic structures effectively (Choi et al., 2022b). We implement the application of the analogous operation to temporal traffic data by utilizing the clique adjacency matrix $A_C$, the incorporation of which is equivalent to connecting nodes representing traffic at different instances of time with edges. In other words, the clique adjacency matrix $A_C$ allows room for GNNs to determine whether additional interactions among nodes of $\mathcal{G}$, whose measurements are taken at varying moments of time, are relevant components for forecasting traffic signals. By doing so, we are able to effectively encapsulate the potential effects of topological invariants of traffic networks on temporal traffic measurements, observation of which was inferred from Theorem 3.1. Figure 1 illustrates how lifting a cyclic subgraph of $\mathcal{G}$ to a new cyclic subgraph of $\mathcal{G} \times I$ of varying temporal instances can enrich our prediction of future traffic signals. The two cycles are subgraphs colored in red. Note that in this example, the temporal cyclic subgraph spans over time $t = t_0$ and the end time $t = t_e$ of the dataset. By changing the colored cyclic subgraph into cliques, we add edges to nodes on the cyclic subgraph lying in two different instances of time. These edges establish relations among measurements taken at different nodes and time periods.

# 4 METHODOLOGY

In this section, we present the architecture of Cy2Mixer and elaborate on how it distinguishes itself from other models. Figure 2 provides a comprehensive visualization of our model's framework. This architecture is primarily composed of three main components: the embedding layer, a stack of $L$ Cy2Mixer encoder layers, and the output layer. Further details on these components are elaborated upon subsequently.

## 4.1 EMBEDDING LAYER

Our model follows the structure of the embedding layer from Liu et al. (2023). Given the time-series data $X_{t-T+1:t}$, the feature embedding is defined as $X_{feat} = \text{FC}(X_{t-T+1:t})$, where FC represents a fully connected layer and $X_{feat} \in \mathbb{R}^{T \times N \times d_f}$. Additionally, to account for both weekly and daily periodicities, the temporal embedding $X_{temp} \in \mathbb{R}^{T \times N \times d_t}$ is produced. This is achieved by referencing the day-of-week and timestamp-of-day data to extract two distinct embeddings, which are subsequently concatenated and broadcasted to generate the final temporal embedding with dimension $d_t$. Furthermore, to address diverse temporal patterns specific to each node, an adaptive embedding of dimension $d_a$, represented as $X_{ast} \in \mathbb{R}^{T \times N \times d_a}$, is introduced to capture diverse temporal patterns for each node. These three embeddings are concatenated, forming the output $H \in \mathbb{R}^{T \times N \times d_h}$ of the embedding layer:

$$H = X_{feat} || X_{temp} || X_{ast},$$

where the dimension $d_h$ is equal to $d_f + d_t + d_a$.

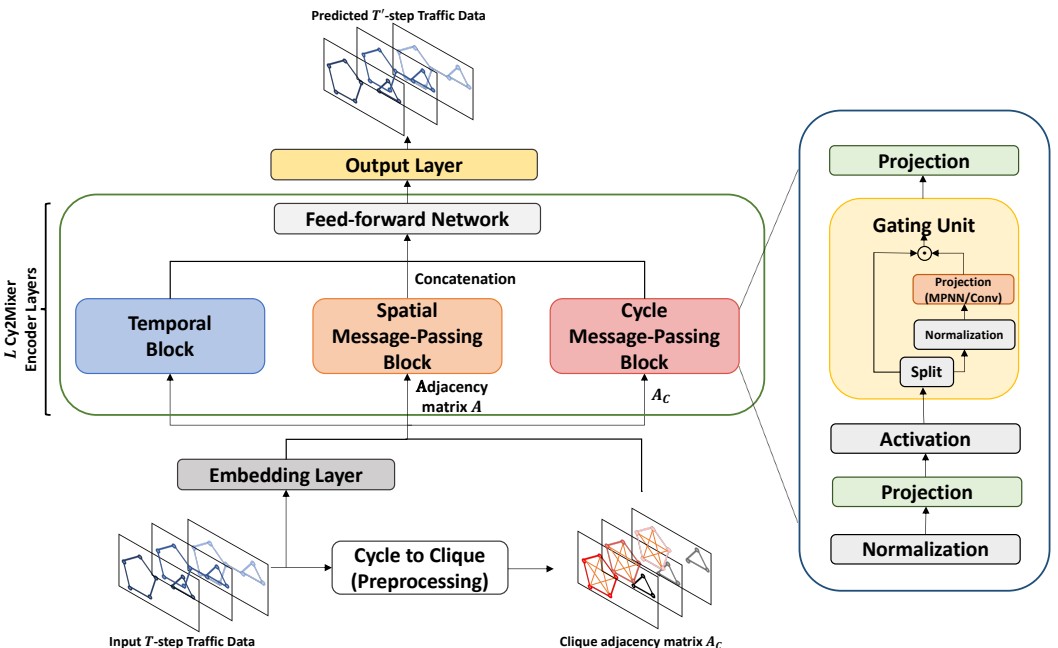

Figure 2: The overall framework of Cy2Mixer. Layers within the green box indicate the Cy2Mixer encoder layer, which comprises a temporal block, spatial message-passing block, and cycle message-passing block to ensure a comprehensive understanding of both temporal and spatial aspects.

## 4.2 CY2MIXER ENCODER LAYER

The core architecture of the Cy2Mixer encoder layer consists of three distinct blocks, with each calculating the projection values of $Z$ based on the output $H$ from the embedding layer. Upon splitting $Z \in \mathbb{R}^{T \times N \times 2d_h}$ into $Z_1 \in \mathbb{R}^{T \times N \times d_h}$ and $Z_2 \in \mathbb{R}^{T \times N \times d_h}$, each block adopts its unique methodology within the Gating Unit.

**Temporal Block** This block employs a $3 \times 3$ convolution network to get the projection values of $Z_2$. This is subsequently element-wise multiplied with $Z_1$, generating the output of the Gating Unit:

$$\tilde{Z}_{temporal} = Z_{temporal,1} \odot \text{Conv}(Z_{temporal,2}), \quad \tilde{Z}_{temporal} \in \mathbb{R}^{T \times N \times d_h},$$

where $Z_{temporal,1}$ and $Z_{temporal,2}$ represents $Z_1$ and $Z_2$ for the Temporal block, respectively.

**Spatial Message-Passing & Cycle Message-Passing Blocks** These blocks employ the MPNN for their projection function in the Gating Unit. Notably, the Spatial Message-Passing block uses the standard adjacency matrix, $A$, whereas the Cycle Message-Passing block operates with the clique adjacency matrix, $A_C$. The clique adjacency matrix, proposed by Choi et al. (2022b), represents the bases of cycles (or the first homological invariants of the graph in a graph (Paton, 1969)) in a suitable form that enables GNNs to effectively process the desired topological features. Following a process similar to the Temporal block, the $\tilde{Z}$ values for both blocks are expressed as:

$$\tilde{Z}_{spatial} = Z_{spatial,1} \odot \text{MPNN}(Z_{spatial,2}, A), \quad \tilde{Z}_{temporal} \in \mathbb{R}^{T \times N \times d_h}$$

$$\tilde{Z}_{cycle} = Z_{cycle,1} \odot \text{MPNN}(Z_{cycle,2}, A_C), \quad \tilde{Z}_{temporal} \in \mathbb{R}^{T \times N \times d_h}.$$

The final output of each block, denoted as $Y$, is calculated using the output of the Gating Unit, $\tilde{Z}$. Considering the effective performance outcomes of incorporating tiny attention into each block in prior research (Liu et al., 2021), we have adopted the same structure in our model. Then we concatenate all of three outputs, namely, $Y_{temporal}, Y_{cycle}$, and $Y_{spatial}$, using a feed-forward network as:

$$Y_{out} = \text{FC}\left(Y_{temporal} || Y_{cycle} || Y_{spatial}\right), \quad Y_{out} \in \mathbb{R}^{T \times N \times d_h}$$

## 4.3 OUTPUT LAYER

After progressing through the stacked sequence of $L$ Cy2Mixer encoder layers, the output layer extracts the final predictions from the hidden state $Y_{out}$.

$$\hat{Y} = \text{FC}(Y_{out}), \quad Y \in \mathbb{R}^{T' \times N \times d_o}.$$

Here, $T'$ represents the number of time steps to be predicted, and $d_o$ signifies the dimension of the output features.

## 5 EXPERIMENTS

**Dataset** We evaluate Cy2Mixer's performance using five real-world public traffic datasets. These datasets contain three datasets including only the traffic data, namely PEMS04, PEMS07, and PEMS08 (Song et al., 2020), and three datasets including inflow and outflow data, namely NY-Taxi Liu et al. (2020), CHBike Wang et al. (2021), and TDrive Pan et al. (2019). The first three datasets (PEMS04, PEMS07, PEMS08) follow a predictive modeling scheme where the data from the previous one hour (12-time steps) is utilized to forecast the data for the subsequent one hour (12-time steps), while the remaining three datasets (NYTaxi, CHBike, TDrive) use the previous six steps to predict the next one step. Further details are provided in the Appendix.

**Baseline models** In this study, we evaluate our proposed approach against various established baseline methods in traffic forecasting. We include GNN-based methods such as DCRNN (Li et al., 2017), STGCN (Yu et al., 2017), GWNet (Wu et al., 2019), MTGNN (Wu et al., 2020), STFGNN Pan et al. (2019) and STGNCDE Choi et al. (2022a). We also include STTN Xu et al. (2020), GMAN (Zheng et al., 2020), ASTGNN Guo et al. (2021), PDFormer (Jiang et al., 2023) and STAEFormer (Liu et al., 2023), all of which are self-attention-based models designed for the same task as ours.

**Experimental settings** We configured our experiments using settings consistent with existing approaches to ensure fair comparisons. The data split ratios for training, validation, and testing were set to 6:2:2 for PEMS04, PEMS07, and PEMS08, and 7:1:2 for METR-LA and PEMS-BAY. All experiments were conducted on an NVIDIA A100 GPU and 80GB memory and implemented using Python version 3.10.4 with PyTorch. We performed a hyperparameter search to select the optimal model based on its performance on the validation set. Detailed information about these hyperparameters is available in the Appendix.

For evaluating our model's performance, we utilized three widely recognized metrics for traffic forecasting tasks: Mean Absolute Error (MAE), Root Mean Square Error (RMSE), and Mean Absolute Percentage Error (MAPE). Similar to prior research, we assessed the average performance across all 12 forecasted time steps for the PEMS04, PEMS07, and PEMS08 datasets.

**Results from benchmark datasets** The comparison results between our proposed method and various baselines on traffic datasets can be found in Table 1. Based on the results presented in the table, Cy2Mixer outperforms the baseline models across the majority of datasets. Comparing the results in PEMS04, it can be observed that Cy2Mixer demonstrates notable improvements, with MAE decreasing from 18.22 to 18.14, RMSE improving from 30.18 to 30.02, and MAPE showing an enhancement from 11.98% to 11.93%. In the case of the PEMS07 dataset, the proposed model shows lower performance than STAEFormer compared to other datasets. This is because there are no cyclic subgraphs for the PEMS07 dataset, as observed in the Appendix. This outcome underscores the substantial impact of the cycle message-passing block on the model while demonstrating that the proposed model can compete favorably with other models even without utilizing $A_C$ to cycle message-passing block. The robust performance observed in the results of other datasets also demonstrates that Cy2Mixer performs well regardless of the number of predicted time steps or the number of features.

## 6 ABLATION STUDY

**Effectiveness of cycle message-passing block**

Table 1: Traffic flow prediction results for PEMS04, PEMS07, and PEMS08. All prediction results other than the bold method are cited from available results obtained from pre-existing publications. Highlighted are the top first, second, and third results.

| Datasets | Metric | DCRNN | STGCN | GWNET | MTGNN | STFGNN | STGNCDE | STTN | GMAN | ASTGNN | PDFormer | STAEFormer | Cy2Mixer w/o Cycle block | Cy2Mixer |
|---|---|---|---|---|---|---|---|---|---|---|---|---|---|---|
| PEMS04 | MAE | 22.74 | 21.76 | 19.36 | 19.08 | 19.83 | 19.21 | 19.48 | 19.14 | 18.60 | **18.36** | **18.22** | 18.81 | **18.14** |
| | RMSE | 36.58 | 34.77 | 31.72 | 31.56 | 31.87 | 31.09 | 31.91 | 31.91 | 31.03 | **30.03** | **30.18** | 30.65 | **30.02** |
| | MAPE | 14.75% | 13.87% | 13.30% | 12.96% | 13.02% | 12.77% | 13.63% | 13.19% | 12.63 | **12.00%** | **11.98%** | 12.86% | **11.93%** |
| PEMS07 | MAE | 23.63 | 22.90 | 21.22 | 20.82 | 22.07 | 20.62 | 21.34 | 20.97 | 20.62 | 19.97 | **19.51** | 19.71 | **19.45** |
| | RMSE | 36.51 | 35.44 | 34.12 | 34.09 | 35.81 | 34.04 | 34.59 | 34.10 | 34.02 | 32.95 | **32.60** | 33.02 | **32.89** |
| | MAPE | 12.28% | 11.98% | 9.08% | 9.03% | 9.21% | 8.86% | 9.93% | 9.05% | 8.86% | 8.55% | **8.01%** | 8.16% | **8.11%** |
| PEMS08 | MAE | 18.19 | 17.84 | 15.06 | 15.40 | 16.64 | 15.46 | 15.48 | 15.31 | 14.97 | **13.58** | **13.46** | 13.71 | **13.53** |
| | RMSE | 28.18 | 27.12 | 24.86 | 24.93 | 26.21 | 24.81 | 24.97 | 24.92 | 24.71 | **23.41** | **23.25** | 23.63 | **23.22** |
| | MAPE | 11.23% | 11.21% | 9.51% | 10.17% | 10.55% | 9.92% | 10.34% | 10.13% | 9.49% | 9.05% | **8.88%** | 9.01% | **8.86%** |
| NYTaxi | MAE | 13.63 | 13.46 | 13.30 | 13.23 | 14.26 | 13.28 | 13.37 | 13.27 | 12.98 | **12.36** | **12.61** | 12.61 | **12.59** |
| | RMSE | 21.97 | 21.91 | 21.71 | 21.61 | 23.87 | 21.68 | 21.84 | 21.66 | 21.19 | **20.18** | 20.91 | **20.53** | **20.45** |
| | MAPE | 14.35% | 14.16% | 13.94% | 14.73% | 13.93% | 13.98% | 13.65% | 13.98% | 13.65% | **12.79%** | 13.06% | **13.06%** | **13.03%** |
| TDrive | MAE | 21.94 | 21.14 | 19.55 | 18.96 | 22.51 | 19.29 | 20.51 | 19.10 | **18.79** | 17.79 | **16.97** | 17.48 | **16.99** |
| | RMSE | 38.41 | 37.84 | 36.18 | 35.69 | 40.55 | 36.12 | 37.14 | 36.05 | 33.93 | 31.55 | **31.02** | 31.31 | **30.82** |
| | MAPE | 17.57% | 17.26% | 16.56% | 16.41% | 18.54% | 16.50% | 16.66% | 16.45% | 15.84% | 14.68% | **13.81%** | 13.95% | **13.56%** |
| CHBike | MAE | 4.22 | 4.18 | 4.13 | 4.10 | 4.25 | 4.11 | 4.14 | 4.10 | **4.02** | **3.89** | 4.03 | **3.89** | 3.80 |
| | RMSE | 5.91 | 5.87 | 5.81 | 5.74 | 5.90 | 5.80 | 5.83 | 5.79 | 5.71 | **5.48** | 5.70 | **5.46** | 5.37 |
| | MAPE | 31.04% | 31.00% | 30.92% | 30.86% | 32.27% | 30.87% | 31.00% | 30.91% | 30.91% | **30.06%** | 31.49% | **30.10%** | 29.20% |

The design of the cycle message passing block is crucial since it needs to provide additional topological information within the network. We performed an ablation study to evaluate the influence of the cycle message-passing block more comprehensively. Three models were compared: the results of Cy2Mixer's cycle message-passing block using the clique adjacency matrix $A_C$, the results when employing the DTW matrix previously utilized in PDFormer (Jiang et al., 2023) instead of $A_C$ to capture long-range relationships (w/ DTW), and the results where the cycle message-passing block is not used (w/o Cycle block), as presented in Table 2. PEMS04, PEMS07, and PEMS08 were chosen as the datasets for the ablation study due to their varying numbers of nodes and time steps.

Table 2: Ablation study on effect of cycle message-passing block for PEMS04, PEMS07, and PEMS08. Note that w/ stands for with and w/o stands for without.

| Dataset | PEMS04 | | | PEMS07 | | | PEMS08 | | |
|---|---|---|---|---|---|---|---|---|---|
| Metric | MAE | RMSE | MAPE | MAE | RMSE | MAPE | MAE | RMSE | MAPE |
| w/o Cycle block | 18.81 | 30.65 | 12.86% | 19.74 | 33.46 | 8.19% | 13.56 | 23.45 | 8.97% |
| w/ DTW | 18.44 | 30.66 | 12.16% | 19.72 | 33.35 | 8.31% | 13.65 | 23.50 | 8.94% |
| **Cy2Mixer** | **18.14** | **30.02** | **11.93%** | **19.50** | **33.28** | **8.19%** | **13.53** | **23.22** | **8.86%** |

These results indicate that even when both the DTW matrix and $A_C$ are not utilized, the proposed model demonstrates relatively respectable performance when compared to existing models. However, it is evident that the highest performance is achieved when $A_C$ is employed in the cycle message-passing block. Notably, under the same conditions, generating the DTW matrix for 307 nodes took over 3 hours, while creating the clique adjacency matrix took less than a minute. This significant difference in computation time can be attributed to the fact that the DTW method relies on time-dependent similarity calculations for each node, whereas the clique adjacency matrix construction does not necessitate considering all time steps. Based on this result, we can observe that the mathematical proofs we presented indeed have a significant impact on the model.

**Case study** We conducted a case study to analyze the influence of each component of Cy2Mixer by comparing it to variants of the proposed model for PEMS04. First, to demonstrate the effectiveness of our proposed Cy2Mixer block, we compared cases where only the temporal block was used and cases where only the spatial message-passing block was used. In this comparison, the self-attention layer, designed based on the STAEFormer model Liu et al. (2023), replaced the omitted block. Additionally, for the spatial message-passing block and cycle message-passing block, we compared models that either do not use adjacency matrix $A$ and clique adjacency matrix $A_C$ or use them. Next, we conducted experiments by comparing models in which either the spatial message-passing block or the cycle message-passing block was excluded from the structure of Cy2Mixer. Finally, to demonstrate the effectiveness of tiny attention, we compared results with models that did not use tiny attention. The results of these experiments are summarized in the Table 3.

Not only did each component significantly impact the model's performance, but there was also a substantial performance improvement when topological information was incorporated into both the spatial message-passing block and the cycle message-passing block through the adjacency matrix

Table 3: Case study on the structure of Cy2Mixer for PeMS04. Spatial block and Cycle block refer to the spatial message-passing block and cycle message-passing block, respectively, while w/o stands for without.

| PEMS04 | MAE | RMSE | MAPE |
|---|---|---|---|
| only Temporal block | 18.55 | 30.31 | 12.20% |
| only Spatial block (not using $A$ and $A_C$) | 22.53 | 36.33 | 15.36% |
| only Spatial and Cycle block (using $A$ and $A_C$) | 18.56 | 30.33 | 12.32% |
| w/o Spatial block | 18.85 | 30.68 | 12.72% |
| w/o Cycle block | 18.81 | 30.65 | 12.86% |
| w/o Tiny attention | 18.36 | 30.15 | 12.05% |
| **Cy2Mixer** | **18.14** | **30.02** | **11.93%** |

and the clique adjacency matrix. This is evident from the significant reduction in MAE, RMSE, and MAPE when $A$ and $A_C$ were omitted in the spatial message-passing block (the second result) compared to when they were included (the third result). This indicates that in Cy2Mixer, these two matrices exert significant influence on the model's performance.

**Additional experiments for different spatio-temporal learning tasks.**

We conducted additional experiments to validate the proposed Cy2Mixer not only for traffic forecasting but also for various other spatio-temporal tasks. Among these tasks, we specifically verified its performance in the context of air pollution prediction. A whole 4-year dataset *KnowAir* was used for predicting particles smaller than $2.5\mu m$ ($PM_{2.5}$) concentrations (Wang et al., 2020a). Dataset covers in total 184 cities, which can be expressed as nodes, and the dataset is split into 3 sub-datasets based on dates. Since the compared models in this context differ from the models used in the main text, Cy2Mixer was compared with models designed for air pollution prediction; GRU, GC-LSTM (Qi et al., 2019), and $PM_{2.5}$-GNN (Wang et al., 2020a). The results of this comparison were evaluated using root mean squre error (RMSE) and critical success index (CSI), which are commonly used meteorological metrics, and the results are presented in Table 4 .

Table 4: Air pollution prediction results for *KnowAir* datasets. All prediction results other than the bold method are cited from available results obtained from pre-existing publications. Highlited are the top first, second, and third results.

| Dataset | Sub-dataset 1 | | Sub-dataset 2 | | Sub-dataset 3 | |
|---|---|---|---|---|---|---|
| Metric | RMSE | CSI(%) | RMSE | CSI(%) | RMSE | CSI(%) |
| GRU | 21.00 ± 0.17 | 45.38 ± 0.52 | 32.59 ± 0.16 | 51.07 ± 0.81 | 45.25 ± 0.85 | 59.40 ± 0.01 |
| GC-LSTM | 20.84 ± 0.11 | 45.83 ± 0.43 | 32.10 ± 0.29 | 51.24 ± 0.13 | 45.01 ± 0.81 | 60.58 ± 0.14 |
| $PM_{2.5}$-GNN | 19.93 ± 0.11 | 48.52 ± 0.48 | 31.37 ± 0.34 | 52.33 ± 1.06 | 43.29 ± 0.79 | 61.91 ± 0.78 |
| $PM_{2.5}$-GNN no PBL | 20.46 ± 0.18 | 47.43 ± 0.37 | 32.44 ± 0.36 | 51.05 ± 1.15 | 44.71 ± 1.02 | 60.64 ± 0.84 |
| $PM_{2.5}$-GNN no export | 20.54 ± 0.16 | 45.73 ± 0.58 | 31.91 ± 0.32 | 51.54 ± 1.27 | 43.72 ± 1.03 | 61.52 ± 0.95 |
| Cy2Mixer no Cycle block | 19.76 ± 0.13 | 47.95 ± 0.33 | 31.31 ± 0.20 | 52.03 ± 0.95 | 43.27 ± 0.41 | 61.65 ± 0.52 |
| **Cy2Mixer** | **19.34 ± 0.13** | **48.58 ± 0.56** | **31.29 ± 0.19** | 51.64 ± 0.90 | **43.19 ± 0.98** | 62.06 ± 0.69 |

Cy2Mixer consistently demonstrated the highest predictive performance across most datasets. Notably, it achieved this without relying on specific domain knowledge, distinguishing it from the previous $PM_{2.5}$-GNN model. This versatility suggests that the proposed Cy2Mixer can be applied to various spatio-temporal tasks beyond traffic prediction with ease.

## 7 CONCLUSION

In this paper, we have mathematically investigated the effects of topological non-trivial invariants on capturing the complex dependencies of spatio-temporal graphs. Through our investigation, we gained insight into how the homotopy invariance of fundamental groups and homology groups can be a contributing factor in influencing the predictive performance of spatio-temporal graphs. We then introduce a simple yet novel model, Cy2Mixer, based on the mathematical background and inspired by gMLP. Cy2Mixer comprises three major components: a temporal block, a spa-

tial message-passing block, and a cycle message-passing block. Notably, the cycle message-passing block enriches topological information in each of the Cy2Mixer encoder layers, drawing from cyclic subgraphs. Indeed, Cy2Mixer achieves state-of-the-art or second-best performance on traffic forecast benchmark datasets. We further investigate the effects of the cycle message-passing block on benchmark datasets to compare the DTW method, which makes a new adjacency based on the time-dependent similarity for each node. Compared to the DTW method, the cycle message-passing block captures the spatio-temporal dependency more effectively with a significantly lower computational cost. For future work, we plan to extend our study to other spatio-temporal tasks and additional GNN tasks, such as link prediction, by integrating self-supervised techniques to enhance predictive performance compared to other state-of-the-art studies.

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

## A   APPENDIX

### Dataset description

A detail of statistical information of six traffic datasets is outlined in Table 5. The first three datasets are graph-based datasets, each with a single feature, while the latter three are grid-based datasets with two features. The node counts for the grid-based datasets are 75 (15×5), 270 (15×18), and 1,024 (32×32), respectively. The term "# Cycles" indicates the number of the cycle bases of graphs, and the term "Average Magnitude # Cycles" denotes the average number of nodes present in a cycle subgraph of a graph.

Table 5: Description of statistical information of six traffic datasets.

| Datasets | # Nodes | # Edges | # Timesteps | # Time Interval | Time range | # Features | # Cycles | Average Magnitude # Cycles |
|---|---|---|---|---|---|---|---|---|
| PEMS04 | 307 | 340 | 16,992 | 5 min | 01/01/2018-02/28/2018 | 1 | 45 | 4.9111 |
| PEMS07 | 883 | 866 | 28,224 | 5 min | 05/01/2017-08/31/2017 | 1 | 0 | 0 |
| PEMS08 | 170 | 295 | 17,856 | 5 min | 07/01/2016-08/31/2016 | 1 | 105 | 7.8571 |
| NYCTaxi | 75 | 484 | 17,520 | 30 min | 01/01/2014-12/31/2014 | 2 | 168 | 4.8810 |
| TDrive | 270 | 1,966 | 4,416 | 30 min | 07/01/2020-09/30/2020 | 2 | 2883 | 14.4707 |
| CHBike | 1,024 | 7,812 | 3,600 | 60 min | 02/01/2015-06/30/2015 | 2 | 714 | 7.9174 |

### Hyperparameter search

We conducted a hyperparameter search to find the model, and hyperparameters for each dataset are listed in Table 6. Note that the embedding dimensions, $d_f$, $d_t$, and $d_a$, followed the settings of previous research Liu et al. (2023). The search ranges were $\{16, 32, 64, 128\}$ for hidden dimension $d_h$, $\{2, 3, 4, 5, 6\}$ for number of layers, and $\{0, 0.2, 0.4, 0.6, 0.8\}$ for dropout rate, respectively. Considering previous research has demonstrated the effectiveness of incorporating tiny attention, we conducted experiments in our study to compare the performance when tiny attention is added and when it is not. The selection of the optimal model was based on its performance on the validation set. We perform the experiments on STAEFormer (Liu et al., 2023) framework.

Table 6: Hyperparameters for six traffic datasets.

| Datasets | # Layers | $d_f$ | $d_t$ | $d_a$ | $d_h$ | Batch size | Dropout | Weight decay | Learning rate | Learning rate decay | Tiny attention |
|---|---|---|---|---|---|---|---|---|---|---|---|
| PEMS04 | 3 | 24 | 24 | 80 | 152 | 16 | 0.4 | 0.0005 | 0.001 | 0.1 | O |
| PEMS07 | 4 | 24 | 24 | 80 | 152 | 16 | 0.4 | 0.001 | 0.001 | 0.1 | X |
| PEMS08 | 3 | 24 | 24 | 80 | 152 | 16 | 0.1 | 0.0015 | 0.001 | 0.1 | O |
| NYCTaxi | 5 | 24 | 24 | 80 | 256 | 16 | 0.4 | 0.05 | 0.001 | 0.1 | X |
| TDrive | 6 | 24 | 24 | 80 | 256 | 16 | 0.4 | 0.05 | 0.001 | 0.1 | X |
| CHBike | 3 | 24 | 24 | 80 | 256 | 16 | 0.4 | 0.05 | 0.001 | 0.1 | O |

### Comparison of the adjacency matrix, clique adjacency matrix, and DTW matrix.

In this section, a detailed comparison is conducted between the adjacency matrix and clique adjacency matrix used in Cy2Mixer and the DTW matrix employed in PDFormer Jiang et al. (2023).

Three matrices that can be constructed from the traffic network, namely the adjacency matrix $A$, the clique adjacency matrix $A_C$, and the matrix derived from DTW, can be found in Fig. 3. DTW algorithm is a similarity measure between two time-series that allows for non-linear alignment of the time series. In the context of PDFormer, DTW is used to compute the similarity of the historical traffic flow between nodes to identify the semantic neighbors of each node by calculating a pairwise distance matrix encompassing all combinations of historical traffic flow sequences. This distance matrix aids in the computation of a cumulative distance matrix, representing the minimum distance between two time-series up to a specific point. Finally, the cumulative distance matrix facilitates the determination of the optimal warping path, signifying the most favorable alignment between two time-series. DTW algorithm allows the model to capture complex traffic patterns that are not easily captured by linear alignment methods, but it has a significant drawback: it is computationally time-consuming to compute and often relies on heuristic thresholds and parameters. To overcome these limitations and capture the desired spatio-temporal dependencies, we employ the use of the clique adjacency matrix. The clique adjacency matrix can be computed more efficiently than DTW since it is not dependent on time. Furthermore, it provides the model with richer topological information within the graph.

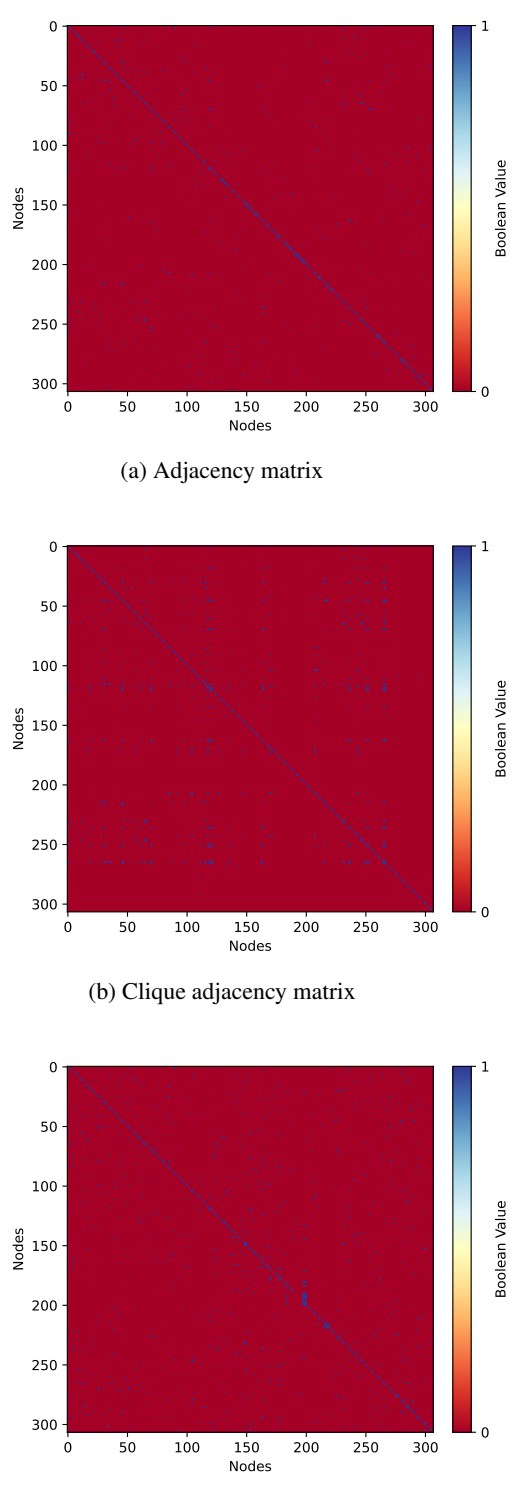

(a) Adjacency matrix

(b) Clique adjacency matrix

(c) Matrix constructed by DTW algorithm

Figure 3: Three matrices of PEMS04: (a) Adjacency matrix $A$, (b) Clique adjacency matrix $A_C$, and (c) Matrix constructed by DTW algorithm, respectively.

