# OpenReview forum: "Pay attention to cycle for spatio-temporal graph neural network"
_ICLR.cc/2024/Conference — Submitted to ICLR 2024_

### Official Review · Reviewer_MKsg · 2023-10-26

**Soundness:** 2 fair
**Presentation:** 2 fair
**Contribution:** 2 fair
**Rating:** 5
**Confidence:** 4

**Summary:**

This work introduces a novel spatio-temporal GNN based model, Cy2Mixer, to extract topological non-trivial invariants of spatio-temporal graphs with gated multi-layer perceptrons (gMLP). The Cy2Mixer is composed of three blocks, i.e., a message-passing block for encapsulating spatial information, cycle message-passing block for enriching topological information through cyclic subgraphs, and a temporal block for capturing temporal properties. Extensive experiments validate the good performances of proposed Cy2Mixer.

**Strengths:**

S1. This paper is well-organized with good mathematical definitions, backgrounds, and an intuitive overview of proposed framework.

S2. The experiments on six datasets are extensive and convincing, along with sufficient ablation studies and case studies. Most experimental results demonstrate the improvement of proposed Cy2Mixer.

S3. The idea of temporal clique is a bit new to the community of spatiotemporal learning.

**Weaknesses:**

W1. Although this paper clarifies that they propose to identify the topological non-trivial invariants in graphs, they have not specified how to find out the invariant graph with temporal dimension.

W2. As this work introduces the temporal cyclic structures, the authors still do not demonstrate how to identify the temporal cycled clique in the spatiotemporal graph.

W3. The title does not match well with this study. Actually, spatiotemporal learning consists of plenty of tasks, which is not limited to traffic forecasting. To name a few, dynamic graph learning for both electricity and air quality forecasting in [1],[2],[3].

W4. The technical contribution is limited. The proposed solution comprises of temporal block, spatial message-passing and a cycle message-passing where the cycled clique is realized with a Clique adjacency matrix 𝑨c (a newly defined adjacent matrix). To this end, first, the three blocks are trivial as usual work, while second, the new solution is inherently a new design of 3D-adjacent matrix. For techniques, the construction process of Ac, how to identify the temporal clique (a temporal cyclic subgraph of the topological space) are lacking, and how the new adjacency contributes to final results is still unclear. More details should be clarified and added.

W5. Typos in this work. eg., ‘colons (:)’ in abstract should be ‘,’.

[1] Wu Z, Pan S, Long G, et al. Connecting the dots: Multivariate time series forecasting with graph neural networks[C]//Proceedings of the 26th ACM SIGKDD international conference on knowledge discovery & data mining. 2020: 753-763.

[2] Zhou Z, Huang Q, Lin G, Yang K, et al. GReTo: Remedying dynamic graph topology-task discordance via target homophily[C]//The Eleventh International Conference on Learning Representations. 2022.

[3] Liang Y, Xia Y, Ke S, et al. Airformer: Predicting nationwide air quality in china with transformers[C]//Proceedings of the AAAI Conference on Artificial Intelligence. 2023, 37(12): 14329-14337.

**Questions:**

Please refer to W1-W4.

---

> ### Author Response · Authors · 2023-11-21
> **Response to Reviewer MKsg**
>
> We thank the reviewer for suggesting various constructive comments. Please find our response to the weaknesses proposed by the reviewer.
>
> - Weakness 1: Theorem 3.1 demonstrates that any cyclic subgraphs of the temporal traffic signal $\mathcal{G} \times I$ is comprised of nodes of form $(c_1, t_1), (c_2, t_2), \cdots, (c_n, t_n) \in \mathcal{G} \times I$, where the nodes $c_1, c_2, \cdots, c_n$ are on a common cyclic subgraph of the underlying traffic network $\mathcal{G}$. We added the above comment after Theorem 3.1 of the updated manuscript.
> - Weakness 2: Using the notation mentioned in our response above, the temporal cycled clique in the spatiotemporal graph corresponds to a complete subgraph with nodes of form $(c_1, t_1), (c_2, t_2), \cdots, (c_n, t_n) \in \mathcal{G} \times I$, where the nodes $c_1, c_2, \cdots, c_n$ are on a common cyclic subgraph of the underlying traffic network $\mathcal{G}$. That is, all the nodes $(c_1, t_1), (c_2, t_2), \cdots, (c_n, t_n)$ are connected by edges by one another. For example, consider a cyclic traffic network consisting of 5 destinations A, B, C, D, and E where the edges correspond to roads connecting destinations A-B, B-C, C-D, D-E, and E-A. A temporal cycle would for example correspond to a graph with nodes $(A,t_1), (B,t_2), (C,t_3), (D,t_2), (E,t_3)$. By using a clique adjacency matrix, which corresponds to connecting all the destination with edges, one is able to understand traffic flow among destinations at different periods of time which do not directly share a road connection, such as understanding traffic flow from destination B at time $t = t_2$ to destination E at time $t = t_3$. More specific details on constructing clique adjacency matrices can be found in Choi et al. (2023).
> - Weakness 3: As the reviewer suggested, we demonstrated the extendability of the proposed model to other spatio-temporal datasets by adding a new experimental result on predicting air quality using KnowAir Dataset. We used this dataset instead of the Airformer dataset as suggested by the reviewer, because the explicit adjacency matrix was not given in the latter dataset, which made us difficult to analyze Airformer dataset using Cy2Mixer given the limited time in writing the response for reviewers. Experimental results suggest that Cy2Mixer outperforms other baseline and state-of-the-art models, thereby suggesting that our proposed model can be used for effectively analyzing other types of spatio-temporal datasets.
> - Weakness 4: As the reviewer rightfully pointed out, it is important to ask why utilizing cyclic structures are helpful to traffic forecasting problems. The significance lies in elucidating the restrictions or constraints associated to constructing a global traffic signal throughout any time period $X: \mathcal{G} \times I \to \mathbb{R}^{N x C}$ from glueing a collection of traffic signals $X_t \in \mathbb{R}^{N x C}$ obtained from a set of distinct time periods. This fact originates from a mathematical insight from algebraic topology and algebraic geometry that given a topological space Y and a collection of open neighborhoods $U_\alpha$ which satisfies $\cup_\alpha U_\alpha = Y$, the constraints associated to glueing continuous functions $f_\alpha: U_\alpha \to \mathbb{R}^k$ to a unique global function $f: Y \to \mathbb{R}^k$ can be understood from computing topological (or cohomological) invariants of a topological space $Y$. In the context of traffic network datasets, the topological space $Y$ corresponds to the temporal traffic network $\mathcal{G} \times I$, the open neighborhoods correspond to traffic network at a certain time frame $\mathcal{G} \times (t_e - 1, t_e + 1)$, the functions $f_\alpha$ correspond to a constant extension of traffic signals $X_{t_e}: \mathcal{G} \times \{t_e\} \to \mathbb{R}^{N \times C}$ to the space $\mathcal{G} \times (t_e - 1, t_e + 1)$, the function $f$ corresponds to the global traffic signal $X: \mathcal{G} \times I \to \mathbb{R}^{N \times C}$, and the obstructions correspond to cyclic subgraphs of $\mathcal{G} \times I$, namely the temporal interactions among nodes of a traffic network which lie on an identical cyclic subgraph. Theorem 3.1 allows us to reduce the problem of computing cyclic subgraphs of $\mathcal{G} \times I$ to computing cyclic subgraphs of $\mathcal{G}$, which implies that previous techniques which analyzes cyclic subgraphs of any graphs can be also utilized to analyze cyclic subgraphs of traffic datasets. We made clarifications addressing the comments above after Theorem 3.1 in the updated manuscript.
> - Weakness 5: As the reviewer suggested, we edited the abstract of the manuscript.

---

### Official Review · Reviewer_LrWr · 2023-10-31

**Soundness:** 3 good
**Presentation:** 1 poor
**Contribution:** 2 fair
**Rating:** 3
**Confidence:** 4

**Summary:**

This paper studies the problem of traffic forecasting and proposes a new design of spatio-temporal graph neural networks to improve performance. Specifically, an extra "clique adjacency matrix" is introduced in the framework to further explore the spatial connections between nodes across different time steps. Comprehensive empirical comparisons with other baseline methods are included in the paper to validate the performance improvement of the proposed method, and codes are provided as supplementary files.

**Strengths:**

1. The empirical comparisons presented in the paper with other $11$ baseline methods, under $3$ metrics, and over $6$ benchmark datasets are fairly comprehensive and extensive. The results show that the proposed method Cy2Mixer can outperform or on par with other sota methods designed for traffic forecasting tasks.

**Weaknesses:**

In summary, I think the problem itself is very interesting, but the quality of this paper is not ready for publication yet. The main weaknesses of this paper are listed as follows:

1. The contribution of this paper is not clear. For the theory part, Theorem 3.1 is a direct result/derivation from a corollary in [1]. Moreover, the connection between the theory and the proposed method is not explicitly shown in the paper. In other words, the theory does not bring any theoretical insights to the readers to understand the proposed method. For the methodology part, the clique adjacency matrix is proposed in [2], the embedding is proposed in [3], and the gated MLP is proposed in [4]. None of these major components within the proposed computing framework is new. Maybe stacking them together can lead to better empirical performance on some datasets, but it seems to be not enough for a publication, unless we have a deeper understanding of why this combination creates better results. I do not think the theory presented in the paper answers this question, at least not in its current form.

2. It is hard to interpret the real contribution of the extra clique adjacency matrix based on the ablation studies. We do not know if the performance improvement comes from extra parameters (since an extra new block is added), or the newly added/rewired edges in the clique adjacency matrix. Moreover, we do not know if the clique adjacency matrix is the best way to add/rewire the edges regarding the performance. For example, spectrum-based rewiring approaches [5, 6] have been widely adopted to resolve the oversquashing issue for graph classification tasks. So I believe more discussions are needed to claim the importance of the usage of clique adjacency matrix.

3. The writing of this paper can be improved. Some sentences do not read well, and some notations are not explained. For example, on page 2, $Z=\sigma(HU)$, what is $H$ here? Since the input is $X$, should it be $Z=\sigma(XU)$? Again on page 2, $s(Z) = Z_1 \odot Proj(Z_2)$, what is exactly the linear projection? Maybe it is better to just keep the original notation which makes more sense, $f_{W,b}(Z)=WZ+b, s(Z)=Z_1 \odot f_{W,b}(Z_2)$. On page 8, what does "tiny attention" refer to? Maybe a reference or a short explanation will help the reader better understand the case study part.

[1] Allen Hatcher. Algebraic Topology. Cambridge University Press, 2002.

[2] Yun Young Choi, Sun Woo Park, Youngho Woo, and U Jin Choi. Cycle to clique (cy2c) graph neural network: A sight to see beyond neighborhood aggregation. In The Eleventh International Conference on Learning Representations, 2022b.

[3] Hangchen Liu, Zheng Dong, Renhe Jiang, Jiewen Deng, Jinliang Deng, Quanjun Chen, and Xuan Song. Spatio-temporal adaptive embedding makes vanilla transformer sota for traffic forecasting. arXiv preprint arXiv:2308.10425, 2023.

[4] Hanxiao Liu, Zihang Dai, David So, and Quoc V Le. Pay attention to mlps. Advances in Neural Information Processing Systems, 34:9204–9215, 2021.

[5] Topping, Jake, Francesco Di Giovanni, Benjamin Paul Chamberlain, Xiaowen Dong, and Michael M. Bronstein. "Understanding over-squashing and bottlenecks on graphs via curvature." arXiv preprint arXiv:2111.14522 (2021).

[6] Karhadkar, Kedar, Pradeep Kr Banerjee, and Guido Montúfar. "FoSR: First-order spectral rewiring for addressing oversquashing in GNNs." arXiv preprint arXiv:2210.11790 (2022).

**Questions:**

Besides the ones listed in the Weaknesses section, I have the following questions:

1. It is mentioned on page 3 that "$A$ and $A_C$ are time-independent input variables since the structure remains unchanged over time". However, in practice, we usually have a (spatial) graph/network that changes dynamically. In the case of long-term traffic forecasting, the changes in the network can be a) some sensors are down due to technical issues; b) rearrangement of the sensors. How would the proposed method change when the graphs can change over time?

2. In intro the authors claim that "the proposed model not only exhibits minimal computational cost but also proves to be highly efficient", but the comparison between the computational costs is not presented in the paper. I think it would be better to add those to the paper as well.

---

> ### Author Response · Authors · 2023-11-21
> **Response to Reviewer LrWr - Part 1**
>
> We sincerely thank the reviewer for providing very detailed comments to our manuscript. Please find our responses to the weaknesses and questions provided by the reviewer.
>
> - Weakness 1: One of the significant contributions of this manuscript lies in verifying that incorporating cyclic substructures of traffic networks (or graph datasets) is effective in forecasting traffic datasets. We note that previous studies which incorporate cyclic substructures of graphs to GNNs, as shown in Choi et al. (ICLR, 2022b), primarily focus on classifying isomorphism classes of collections of different graphs. This is a problem which is of a very different nature to traffic forecasts, which can be understood as a problem focusing on approximating and forecasting collections of functions $f: \mathcal{G} \to \mathbb{R}^{N \times C}$ defined over an underlying traffic network $\mathcal{G}$, or in other words a single underlying graph $\mathcal{G}$. The novelty of the paper hence lies in demonstrating that utilizing cyclic structures of traffic networks is indeed effective in forecasting traffic datasets.
>
>     As the reviewer rightfully pointed out, it is hence a natural question to ask why utilizing cyclic structures are helpful to traffic forecasting problems. The significance of utilizing cyclic substructures of traffic networks lies in elucidating the restrictions or constraints associated to constructing a global traffic signal throughout any time period $X: \mathcal{G} \times I \to \mathbb{R}^{N \times C}$ from glueing a collection of traffic signals $X_t \in \mathbb{R}^{N \times C}$ obtained from a set of distinct time periods. This fact originates from a mathematical insight from algebraic topology and algebraic geometry that given a topological space Y and a collection of open neighborhoods $U_\alpha$ which satisfies $\cup_\alpha U_\alpha = Y$, the constraints associated to glueing continuous functions $f_\alpha: U_\alpha \to \mathbb{R}^k$ to a unique global function $f: Y \to \mathbb{R}^k$ can be understood from computing topological (or cohomological) invariants of a topological space $Y$. In the context of traffic network datasets, the topological space $Y$ corresponds to the temporal traffic network $\mathcal{G} \times I$, the open neighborhoods correspond to traffic network at a certain time frame $\mathcal{G} \times (t_e - 1, t_e + 1)$, the functions $f_\alpha$ correspond to a constant extension of traffic signals $X_{t_e}: \mathcal{G} \times \{t_e\} \to \mathbb{R}^{N \times C}$ to the space $\mathcal{G} \times (t_e - 1, t_e + 1)$, the function $f$ corresponds to the global traffic signal $X: \mathcal{G} \times I \to \mathbb{R}^{N \times C}$, and the obstructions correspond to cyclic subgraphs of $\mathcal{G} \times I$, namely the temporal interactions among nodes of a traffic network which lie on an identical cyclic subgraph. Theorem 3.1 allows us to reduce the problem of computing cyclic subgraphs of $\mathcal{G} \times I$ to computing cyclic subgraphs of $\mathcal{G}$, which implies that previous techniques which analyzes cyclic subgraphs of any graphs can be also utilized to analyze cyclic subgraphs of traffic datasets. We made clarifications addressing the comments above after Theorem 3.1 in the updated manuscript.
>
> - Weakness 2: Unlike the rewiring approach proposed by the reviewer, the clique adjacency matrix can be thought as rewiring the nodes present in a common cyclic subgraph as cliques, as well as eliminating any edges in the underlying graph which do not lie in any of the cyclic subgraphs. By doing so, the proposed model encapsulates the topological non-trivial invariants by incorporating the features obtained from both the usual adjacency matrices and the clique adjacency matrices. In light of the observation that topological non-invariant invariants elucidates the conditions and obstructions associated to gluing traffic signal measurements $X_t$ to obtain a global traffic signal $X$, the proposed method thereby utilizes both types of adjacency matrices to effectively obtain global traffic signals $X$ from adequately gluing traffic signal measurements $X_t$.
> - Weakness 3: As the reviewer kindly pointed out, we made all the relevant clarifications on the notations used in the manuscript.
> - Question 1: If the underlying geometric structure of the traffic network changes, the corresponding adjacency matrix will also change. We believe it is possible to obtain a new clique adjacency matrix based on the newly obtained adjacency matrix, and Cy2Mixer is capable of incorporating the newly obtained clique adjacency matrix and forecast traffic signals. Nevertheless, to the best of our knowledge, the underlying traffic network associated to traffic datasets we used in the experiments are fixed, and no new edges or edges are added to the traffic network with respect to changes in time.

---

> ### Author Response · Authors · 2023-11-21
> **Response to Reviewer LrWr - Part 2**
>
> - Question 2:
>
> | Model | Params | Training time (s) |
> | --- | --- | --- |
> | Cy2Mixer | 2,287,986 | 94.76 |
> | STAEFormer | 1,382,580 | 52.95 |
> | PDFormer | 531,165 | 97.83 |
> | ASTGNN | 473,025 | 82.35 |
>
> We conducted an ablation study which compares the parameters and the computation time required for training respective models. In comparison to other state-of-the-art techniques, Cy2Mixer unfortunately requires using more parameters while exhibiting non-optimal computation time required for training. Nevertheless, as shown in ablation studies in the manuscript, the preprocessing time required to construct clique adjacency matrix is significantly less than the preprocessing time required to construct DTW matrix. For example, as indicated in Page 8 of the manuscript, generating the DTW matrix for a cyclic graph with 307 nodes takes more than 3 hours, whereas generating the clique adjacency matrix takes less than a minute. In fact, while the preprocessing computation cost for generating the DTW matrix depends on the number of nodes of the underlying traffic network and the complexity of time series data, constructing clique adjacency matrices requires less preprocessing time, in particular at most a minute for all the traffic datasets used for empirical results in this manuscript. We believe that because Cy2Mixer can make efficient use of time in preprocessing traffic datasets, our proposed model has potential to efficiently analyze and forecast traffic datasets in comparison to other baseline models.

---

> > ### Comment · Reviewer_LrWr · 2023-11-22
> >
> > I appreciate the efforts made by the authors to prepare the response. However, I am still not fully convinced by the augment in the response to W1 and the new text on page 5. Specifically, why non-trivial invariants can help? Why this choice of non-trivial invariants in simulations? How does it perform compared to some random auxiliary graph? These types of questions could have been explained better by either small toy numerical examples, or maybe some spectral analysis (not sure). I would prefer to maintain my score for now.

---

> > > ### Author Response · Authors · 2023-11-22
> > > **Comment to Reviewer LrWr**
> > >
> > > We thank the reviewer for the follow-up question.
> > >
> > > We particularly chose to use cyclic subgraphs of a traffic network $\mathcal{G}$ instead of random subgraphs of $\mathcal{G}$ because cyclic subgraphs pose additional obstructions required to uniquely glue traffic signals measured over collections of subgraph $\mathcal{H}_i \subset \mathcal{G}$ to the traffic network $\mathcal{G}$.
> > >
> > > Here’s an example we can consider. Suppose we have two graphs $V$ and $U$ without any cycles and continuous functions $f: V \to \mathbb{R}^k$ and $g: U \to \mathbb{R}^k$ defined over the respective graphs. A natural question to ask is if $G$ is a graph such that $G = V \cup U$, what are the conditions required over the functions $f$ and $g$ to guarantee the existence of a global function $F: G \to \mathbb{R}^k$ whose restriction to each subgraphs $V$ and $U$ agree with $f$ and $g$? The answer to this question is that as long as $f$ and $g$ are identical over the intersection of two graphs $V \cap U$, there exists a global function $F: G \to \mathbb{R}^k$ which satisfies the aforementioned condition. The crucial part here, however, is the nature of the topological structure of the intersection $V \cap U$. If it is the case that $G$ does not possess any cyclic subgraphs and $V$ and $U$ are connected, then $V \cap U$ is also connected. In particular, there is a single connected component over which the functions $f$ and $g$ have to agree. This is unfortunately not the case when $G$ contains a cyclic subgraph: it must be the case that $V \cap U$ has at least $2$ connected components, and the functions $f$ and $g$ have to agree on at least two distinct connected components. What this suggests is that the existence of cyclic subgraphs is closely associated to additional components of subgraphs of $G$ where locally defined functions over a collection of subgraphs must agree with each other, in other words additional obstructions imposed to glue locally defined functions to construct a global function. And we would like to note that one of the ways that Cy2Mixer encapsulates such additional obstructions obtained from cyclic subgraphs of $\mathcal{G}$ is by utilizing the cycle message passing blocks.

---

### Official Review · Reviewer_6VA9 · 2023-11-02

**Soundness:** 2 fair
**Presentation:** 2 fair
**Contribution:** 2 fair
**Rating:** 5
**Confidence:** 4

**Summary:**

This paper proposes a novel spatiotemporal graph neural network called Cycle to Mixer (Cy2Mixer) for traffic forecasting. The key contributions are: 1) Provides mathematical analysis using topological invariants to show that cyclic substructures in the traffic network graph can influence predictions. 2) Proposes the Cy2Mixer architecture composed of three main blocks: a temporal block, a spatial message passing block, and a cycle message passing block. 3) Achieves state-of-the-art results on multiple traffic forecasting benchmarks including PEMS04, PEMS07, PEMS08, NYTaxi, TDrive, and CHBike datasets.

**Strengths:**

1) The proposed Cy2Mixer architecture is simple yet effective at integrating spatial, temporal, and topological information through the three blocks.
2) Empirical results surpass prior state-of-the-art methods on multiple traffic forecasting benchmarks.
3) Ablation studies clearly demonstrate the value of the cycle message passing block.

**Weaknesses:**

The writing in this manuscript lacks clarity, and the notation is unclear. For example, on page 2, the meaning of symbols Z1 and Z2 is not well-defined, and k_m does not appear in any formulas. Furthermore, on page 3, the variable 'g' in the equation is not adequately explained.

The theoretical part of this paper is challenging to follow, and it is unclear why the theoretical result (Theorem 3.1) indicates the importance of a cycle basis for the task. I would, therefore, suggest that the authors work on improving the presentation to make the logic behind certain statements more apparent.

Key ablation study is absent from the manuscript. It would be valuable for the author to investigate the effectiveness of the preprocessing step and possibly compare it to other edge-enhancing methods.

**Questions:**

Q1 see the weakness part
Q2 On page 7, authors mentioned the PEMS07 does not have cyclic subgraphs. Can authors clarify how this it is observed?

---

> ### Author Response · Authors · 2023-11-21
> **Response to Reviewer 6VA9**
>
> We thank the reviewer for providing constructive feedback to our manuscript. Please find our response to the weaknesses and questions posed by the reviewer.
>
> - Weakness 1: As the reviewer kindly pointed out, we made all the relevant clarifications on the notations used in the manuscript.
> - Weakness 2: This is a great question. The significance of utilizing cyclic substructures of traffic networks lies in elucidating the restrictions or constraints associated to constructing a global traffic signal throughout any time period $X: \mathcal{G} \times I \to \mathbb{R}^{N \times C}$ from glueing a collection of traffic signals $X_t \in \mathbb{R}^{N \times C}$ obtained from a set of distinct time periods. This fact originates from a mathematical insight from algebraic topology and algebraic geometry that given a topological space Y and a collection of open neighborhoods $U_\alpha$ which satisfies $\cup_\alpha U_\alpha = Y$, the constraints associated to glueing continuous functions $f_\alpha: U_\alpha \to \mathbb{R}^k$ to a unique global function $f: Y \to \mathbb{R}^k$ can be understood from computing topological (or cohomological) invariants of a topological space $Y$. In the context of traffic network datasets, the topological space $Y$ corresponds to the temporal traffic network $\mathcal{G} \times I$, the open neighborhoods correspond to traffic network at a certain time frame $\mathcal{G} \times (t_e - 1, t_e + 1)$, the functions $f_\alpha$ correspond to a constant extension of traffic signals $X_{t_e}: \mathcal{G} \times \{t_e\} \to \mathbb{R}^{N \times C}$ to the space $\mathcal{G} \times (t_e - 1, t_e + 1)$, the function $f$ corresponds to the global traffic signal $X: \mathcal{G} \times I \to \mathbb{R}^{N \times C}$, and the obstructions correspond to cyclic subgraphs of $\mathcal{G} \times I$, namely the temporal interactions among nodes of a traffic network which lie on an identical cyclic subgraph. Theorem 3.1 allows us to reduce the problem of computing cyclic subgraphs of $\mathcal{G} \times I$ to computing cyclic subgraphs of $\mathcal{G}$, which implies that previous techniques which analyzes cyclic subgraphs of any graphs can be also utilized to analyze cyclic subgraphs of traffic datasets. As the author suggested, we made clarifications addressing the comments above after Theorem 3.1 in the updated manuscript.
> - Weakness 3: To ensure a fair and unbiased comparison of Cy2Mixer with other baseline models, we applied the same preprocessing steps as indicated in previous baseline models. The empirical results suggest that under the identical preprocessing protocols, Cy2Mixer still exhibits comparable or outperformance in forecasting traffic dataset in comparison to previous baseline techniques.
> - Question 2: In Table 5, we added a summary of the number of cycles and the average magnitude of cycles for each traffic datasets. As indicated in the original manuscript, PeMS07 do not possess any cyclic subgraphs unlike any other traffic datasets.

---

> > ### Comment · Reviewer_6VA9 · 2023-11-22
> >
> > I thank the authors for their efforts in composing the rebuttal. However, the concerns are not fully solved by the response. Specifically, section 2.2 is still not clear, how are Z1 and Z2 split from Z? If this is a result of a previous work, authors may just hide the details or put it in the appendix. In addition, there are still gaps between the theorem and how it contributes to solving the problem. I would maintain my score for now.

---

> > > ### Author Response · Authors · 2023-11-23
> > > **Follow-up comment to Reviewer 6VA9**
> > >
> > > We thank the reviewer for the follow-up questions.
> > >
> > > We would like to note that the matrix $Z \in \mathbb{R}^{T \times N \times 2d_{h}}$ can be decomposed into $Z_1, Z_2 \in \mathbb{R}^{T \times N \times d_{h}}$, where $T$ is the number of time steps, and $N$ is the number of nodes. Further explanations on these matrices can be found in the reference “Pay Attention to MLPs” by Liu et al., as well as the introduction to Section 4.2 of our manuscript.
> > >
> > > As for the relation between the theorem and contributions to solving the problem, we would like to note that choosing cyclic subgraphs of a traffic network $\mathcal{G}$ allows us to pose additional obstructions required to uniquely glue traffic signals measured over collections of subgraphs $\mathcal{H}_i \subset \mathcal{G}$ to the traffic network $\mathcal{G}$. Here’s an example we can consider. Suppose we have two graphs $V$ and $U$ without any cycles and continuous functions $f: V \to \mathbb{R}^k$ and $g: U \to \mathbb{R}^k$ defined over the respective graphs. A natural question to ask is if $G$ is a graph such that $G = V \cup U$, what are the conditions required over the functions $f$ and $g$ to guarantee the existence of a global function $F: G \to \mathbb{R}^k$ whose restriction to each subgraphs $V$ and $U$ agree with $f$ and $g$? The answer to this question is that as long as $f$ and $g$ are identical over the intersection of two graphs $V \cap U$, there exists a global function $F: G \to \mathbb{R}^k$ which satisfies the aforementioned condition. The crucial part here, however, is the nature of the topological structure of the intersection $V \cap U$. If it is the case that $G$ does not possess any cyclic subgraphs and $V$ and $U$ are connected, then $V \cap U$ is also connected. In particular, there is a single connected component over which the functions $f$ and $g$ have to agree. This is unfortunately not the case when $G$ contains a cyclic subgraph: it must be the case that $V \cap U$ has at least $2$ connected components, and the functions $f$ and $g$ have to agree on at least two distinct connected components. What this suggests is that the existence of cyclic subgraphs is closely associated to additional components of subgraphs of $G$ where locally defined functions over a collection of subgraphs must agree with each other, in other words additional obstructions imposed to glue locally defined functions to construct a global function. And we would like to note that one of the ways that Cy2Mixer encapsulates such additional obstructions obtained from cyclic subgraphs of $\mathcal{G}$ is by utilizing the cycle message passing blocks.

---

### Official Review · Reviewer_VYt1 · 2023-11-04

**Soundness:** 2 fair
**Presentation:** 3 good
**Contribution:** 2 fair
**Rating:** 5
**Confidence:** 3

**Summary:**

This paper focuses on the intricate dependencies in the spatio-temporal traffic graph. Starting from cyclic subgraphs of the traffic network, the paper proposes topological non-trivial invariants based on the homotopy invariance of fundamental groups and homology groups. Consequently, the cyclic subgraphs are transformed into cliques to enhance the relations between different nodes at different timestamps. The CY2MIXER encoder layer is devised to introduce cycle messages-passing blocks. The model is evaluated in various datasets and ablation studies are conducted to verify the effectiveness of each block.

**Strengths:**

1. Unlike other studies on spatio-temporal traffic graphs driven by findings in the transportation system, the theoretical analysis of topological non-trivial invariants is intriguing as it demonstrates how to model the correlations in this scenario.
2. The manuscript is easy to follow and well-organized.
3. The experimental results attain nearly all state-of-the-art or second-best performance across six datasets.

**Weaknesses:**

1. While the paper provides theoretical analysis, the conclusions drawn are not particularly insightful. It's fairly evident that "solely analyzing pre-existing edges cannot address temporal traffic data", a focus of many recent spatio-temporal traffic graph forecasting studies. Additionally, the concept of transforming cyclic subgraphs into cliques in a transportation system isn't clearly explained in real-world scenarios.

2. The model's performance on the PEMS07 dataset is not so satisfactory. The stated reason is "there are no cyclic subgraphs for the PEMS07 dataset", which leads to concerns about the model's limitations; it appears to only handle traffic graphs with cycles. Therefore, it would be beneficial to conduct experiments on traffic graphs with varying proportions of cycles.

3. Despite the results surpassing most baselines, the improvements are marginal. It is unclear whether the same hyperparameter search process was conducted with other baselines. Also, the reasons for only reporting STAEFormer's results on three datasets in Table 1 are not explained.

**Questions:**

See Weakness.

---

> ### Author Response · Authors · 2023-11-21
> **Response to Reviewer VYt1**
>
> We thank the reviewer for providing constructive comments. We would like to address the weaknesses pointed out by the reviewer as follows.
>
> - Weakness 1: One of the key insights utilized in previous studies, such as utilizing the DTW matrix or utilizing dynamically varying adjacency matrices, is that variations in geometric properties of graphs can clarify restraints or conditions that traffic signals at each time period have to satisfy. These geometric conditions are of great importance in deducing global traffic behaviors from conglomerating instances of traffic signals. In this manuscript, we focus on deducing these geometric constraints by extracting cyclic substructures of temporal traffic networks, the problem of which can be reduced to extracting cyclic substructures of pre-existing edges of traffic networks as shown in Theorem 3.1. These cyclic structures form the backbone of topological invariants of temporal traffic networks, the topological properties of which can be effectively used for traffic forecasting.
>
>     As for real-world applications, consider a cyclic traffic network consisting of 5 destinations A, B, C, D, and E where the edges correspond to roads connecting destinations A-B, B-C, C-D, D-E, and E-A. By using a clique adjacency matrix, which corresponds to connecting all the destination with edges, one is able to understand traffic flow among destinations at different periods of time which do not directly share a road connection, such as understanding traffic flow from destination B to destination D.
>
> - Weakness 2: In Table 5, we added a summary of the number of cycles and the average magnitude of cycles for each traffic datasets. As indicated in the table, we can conclude that our original experimental protocols conduct experiments on traffic graphs with varying proportions of cycles.
> - Weakness 3: We utilized identical hyperparameter search processes as suggested in previous studied (in particular PDFormer and STAEFormer), and compared empirical results under identical experimental settings. As for the experimental results obtained from STAEFormer, we added new experimental results obtained from traffic datasets which were not tested in previous studies.

---

### Meta-Review · Area_Chair_meob · 2023-12-06

**Metareview:**

The paper proposes the Cycle to Mixer (Cy2Mixer), which is a new spatiotemporal GNN based on topological non-trivial invariants of spatiotemporal graphs with gated multi-layer perceptrons (gMLP). The proposed Cy2Mixer consists of  three blocks: a message-passing block for encapsulating spatial information, a cycle message-passing block for enriching topological information through cyclic subgraphs, and a temporal block for capturing temporal properties. Experiments have been conducted on  five real-world public traffic datasets to demonstrate the effectiveness of Cy2Mixer.

Reviewers are mostly concerned that the connection between the theory and the proposed method is not clearly established. In addition, the three key building blocks of the proposed Cy2Mixer are based upon existing works as well. While the integration of these block could lead to a sufficiently novel model as a whole, the paper in this current form does not convey a clear enough message to justify that the overall contribution is significant. Multiple reviewers participated in the discussion with the authors during the rebuttal process. However, none of them was fully convinced by the authors' response and decided to keep their original scores in the end.

**Justification For Why Not Higher Score:**

The connection between the theory and the proposed method is not clearly established, making it hard to assess the significance of the overall contribution.

**Justification For Why Not Lower Score:**

N/A

---

### Decision · Program_Chairs · 2024-01-16

Reject